# Diabetes self-care intervention strategies and their effectiveness in Sub-Saharan Africa: A systematic review

Temesgen Anjulo Ageru[1,2,3], Cua Ngoc Le[1,2], Apichai Wattanapisit[4], Eskinder Wolka Woticha[3], Nam Thanh Truong[1,2,5], Muhammad Haroon Stanikzai[1,2,6], Temesgen Lera Abiso[3], Charuai Suwanbamrung[1,2]*

1 Public Health Research Program, School of Public Health, Walailak University, Nakhon Si Thamarata, Thailand, 2 Excellent Center of Dengue and Community Public Health (EC for DACH), Walailak University, Nakhon Si Thamarata, Thailand, 3 Wolaita Sodo University College of Medicine and Health Sciences, Wolaita Sodo, Ethiopia, 4 School of Medicine Walailak University, Nakhon Si Thamarata, Thailand, 5 Faculty of Public Health, Can Tho University of Medicine and Pharmacy, Can Tho city, Vietnam, 6 Faculty of Medicine, Department of Public Health, Kandahar University, Kandahar, Afghanistan

* yincharuai@gmail.com

**Data Availability Statement:** All relevant data are within the paper and its Supporting Information files.

## Abstract

### Background

Diabetes poses a significant public health challenge, predominantly affecting low-and middle-income countries (LMICs), including in Sub-Saharan Africa (SSA). The evolving landscape characterized by resource constraints, gaps in availability and functionality of healthcare system, and socio-demographic impediments has compounded these challenges. As an example, self-care interventions have played a crucial role in diabetes care. However, the effectiveness of these interventions in the SSA remains insufficiently explored. Therefore, this systematic review evaluated the effectiveness and implementation approach of diabetes self-care interventions in SSA.

### Method

A comprehensive search was conducted across electronic databases including PubMed, Scopus, and Google Scholar, to identify studies focusing on diabetes self-care interventions in SSA from 2013 to 2023. The included studies reported interventions targeting dietary adherence, physical activity, medication adherence, blood glucose monitoring, foot care, and educational or support programs. The quality of the studies was assessed using the JBI checklist. Effectiveness was assessed through outcomes including glycemic control, adherence rate, complication reduction, and improvement in patient outcomes.

### Result

Overall, 38.5% of interventions result showed positive changes in either primary or secondary outcomes. Studies that employed diabetes self-management education showed positive changes in approximately 28.6% of cases. However, SMS text messaging interventions did not demonstrate significant changes in the measured outcomes.

**Funding:** This research work was financially supported by Walailak University Graduate Research Fund (No. CGS-RF-03/2023). The funders had no role in study design, data collection and analysis, decision to publish, or preparation of the manuscript.

**Competing interests:** The authors declared there is no conflict of interest.

Of the 13 studies reviewed, 12 used randomized controlled trial designs, whereas one study employed a quasi-experimental approach. The total of participants across intervention and control groups was 3172 adults with diabetes. The interventions employed various strategies including diabetes self-management education led by nurses and other professionals, SMS text messaging for treatment adherence, and other targeted approaches. The duration of these interventions varied from 2 to 12 months. Specific interventions, such as glucose machine provision with training, family support with culturally tailored educations, and periodontal treatment, exhibited notable improvement in adherences and reductions in HbA1c levels.

## Conclusion

The review underscores the significance of diabetes self-care interventions in SSA, showing varying effectiveness levels across different strategies. It emphasizes the importance of tailored approaches and highlight interventions that have shown promising outcomes, providing insights for future research, policy, and healthcare strategies in the region.

## 1. Introduction

Diabetes, defined by elevated blood glucose levels stemming due to impaired insulin secretion or insulin action, is a significant global health concern [1]. Over half a billion individuals are affected worldwide, particularly in low and middle-income countries [2]. Africa faces a growing diabetes burden, with 24 million adults currently affected and projected to increase to 55 million by 2045 [3]. Alarmingly, in 2021 alone, diabetes accounted for half a million deaths across the continent, primarily impacting the most productive segment of society aged 20–59 years in all African regions [3,4].

The chronic nature of diabetes necessitates vigilant self-care to prevent complications that lead to irreversible disability and death if left undiagnosed or inadequately managed [5,6]. Essential self-care aspects outlined in the American Diabetes Association (ADA) and the American Association of Diabetes Care and Education Specialists (ADCES) encompass maintaining a healthy diet, physical activity, blood glucose monitoring, medication adherence, and foot care checks [5,7].

Despite the criticality of self-care, studies in Sub-Saharan Africa revealed concerning adherence rates to recommended management practices. In various countries, low adherence to recommended physical activity level, medications, and blood glucose monitoring has been documented [8–10]. Furthermore, the prevalence of foot ulcers due to inappropriate care among diabetes patients in Africa ranged between 13% and 21.1%, contributing significantly to morbidity and mortality [11–13].

To understand the pivotal role of self-care in diabetes management, various interventions have been proposed and implemented globally, aiming to enhance patient outcomes. These include dietary adherence, medication adherence, regular physical exercise, blood glucose monitoring, foot care, and various educational and support programs [14–16].

Studies have indicated that family support interventions positively impacted diabetes self-care, with patients receiving more family care support demonstrating higher levels of self-care [17,18]. Additionally, diabetes education reinforced by nurses has shown reductions in

HgA1C level, improved lipid profiles, and controlled systolic blood pressure over the medium and long term [19].

Studies in Sub-Saharan Africa showed that text messages to type 2 diabetes patients moderately improved medication adherence but had low impact on physical activity and other practices. These findings highlighted the need to develop effective strategies for enhancing adherence to recommended lifestyle changes in this setting [20,21].

At the global level, systematic reviews on the effectiveness of diabetes self-care care interventions, such as phone-based interventions [22], diabetes self-management education [23,24], and family support [25], hove shown positive effects in controlling glycemic levels and reducing complications among intervention groups.

Despite these global interventions, there is a lack of comprehensive assessment concerning their efficacy in Sub-Saharan Africa. Our systematic review seeks to bridge this gap by evaluating the diabetes self-care interventions implemented in the region, aiming to provide valuable insights for healthcare providers, policymakers, and researchers. To address this gap, the researchers asked questions such as: 1) What are diabetes self-care interventions currently being utilized? 2) How effective are they in Sub-Saharan Africa? The research objective is to identify the interventional strategies for diabetes self-care practices in Sub-Saharan Africa and evaluate their effectiveness employed for diabetes self-care practices in the Sub-Saharan African context.

## 2. Method

### 2.1 Search strategy

A comprehensive search was conducted using various databases, including Pub-Med, Scopus, and Google Scholar of studies published from January 2013 to December 2023. The search strategy incorporated a combination of Medical Subject Headings (MeSH) terms and keywords related to diabetes, self-care, intervention, and Sub-Saharan Africa. Boolean operators (AND, OR) were utilized to effectively combine search terms.

Before conducting the actual search, pilot testing was carried out on PubMed database to evaluate the effectiveness and search terms adjusted. Subsequently, the same approach was applied to other databases.

The search strategy included variations of (("Diabetes" OR "Diabetic" OR "Type 2 diabetes mellitus") AND ("Self-care" OR "Self-management" OR "Lifestyle modification" OR "Patient education" OR "Self-monitoring" OR "Technology-based intervention") AND ("Sub-Saharan Africa" OR "Africa" OR "African countries") AND ("Glycemic control" OR "Quality of life" OR "Complications" OR "Healthcare utilization")).

With registration number (CRD42023451325), the study protocol was registered on PROSPERO, the international perspective register of systematic review.

The research focused on a population comprising individuals with type 2 diabetes residing in Sub-Saharan Africa. The intervention strategies included a comprehensive approach, that incorporated self-care management, lifestyle modification, social support, patient educations, and technology-based interventions. A comparison was made between the intervention and control group, with control receiving usual care or standard care, and intervention group receiving self-care interventions. The study's primary outcome was centered on glycemic control and complication related to diabetes. Glycemic control served as a crucial parameter to evaluate the effectiveness of interventions in managing blood glucose levels. Secondary outcomes measured included improvement in the quality of life among participants and their adherence to self-care practices.

## 2.2 Inclusion and exclusion criteria

Inclusion criteria were designed to focus on studies that match the objectives of the review. First, the study had to be published in English to facilitate comprehension and analysis. Additionally, the target population comprised individuals aged 18 years and above who were diagnosed with type 2 diabetes. Geographically, the studies were limited to those conducted in Sub-Saharan African countries, as defined by the World Bank in 2020 [26]. The interventions were diabetes self-care education, technology based, and any support that used to improve the diabetes self-care activities such diet, glucose monitoring, medication adherence, physical activities, and psychosocial support. Eligible study designs included randomized control trials and quasi-experimental studies. Outcomes included glycemic control, knowledge and behavioral change, adherence, and reduced complications. Lastly, the time frame for inclusions for publication year was set between 2013 and 2023.

Exclusion criteria studies unclearly evaluated outcomes were also excluded. Additionally, studies conducting interventions on type 1 diabetes, or those comparing type 2 diabetes with any other any diseases, were excluded from the review.

Systematic review articles, commentaries, and conference proceedings were excluded to ensure the inclusion of primary research with rigorous methodologies.

## 2.3 Study selection

The Preferred Review Items for Systematic Review and Meta-Analysis (PRISMA) checklist guided the study selection process. Microsoft Excel used to manage articles. Two independent reviewers (T.A & T.L) screened the articles based on predetermined eligibility criteria. Title, abstract, and full text assessments were conducted. Disagreements between the two reviewers were resolved through discussion with third reviewers (T.T.N, M.H.S). Duplicate studies were removed, and included articles focused on diabetes self-care strategies in Sub-Saharan African countries.

## 2.4 Quality and bias assessment

The quality and risk of bias assessment for each included study employed an appropriate tool, the Joanna Briggs Institute JBI [27] checklist for the randomized controlled trials study design, which contains 13 questions (S1 File). For the quasi-experimental design, we used JBI which contains nine questions to assess quality and bias (S2 File). The evaluators responsible for quality and biases assessment held the rank of associate professor and possessed significant experience in the field. Two independent reviewers (CS and CN) assessed the quality of the studies and addressed any discrepancies through consensus with third reviewers (AW, EW). In the checklist, we assigned value of 1 for a 'yes' and 0 for a' no' response. Articles with an overall score of 60 or above out of 100 were considered for inclusion. According to the quality assessment results, the maximum score was 92%, the minimum was 62%, and the average was 77.8% for randomized controlled trial studies (S1 File).

## 2.5 Data extraction and synthesis

The reviewers involved in this study were trained and experienced in systematic review methods. The initial screening of all titles and abstracts was conducted independently by the first reviewer (TA) and the second reviewer (TL) to exclude irrelevant content. Data extraction from selected studies using Microsoft Excel, including information first author, publication year, country, study design, number of participants in intervention and control group,

intervention type, and duration of intervention, effectiveness, and key finding of glycemic control, adherence, and complications (S2 File).

Upon data extracting, the diabetes self-care interventions implemented in each study were identified. The effectiveness was evaluated based on either the primary or secondary outcome measured. The studies with similar interventions were grouped and narrated to draw conclusions and recommendations.

### 2.6 Data analysis

After extracting the data, the data were exported to STATA 14 for analysis, and the findings were summarized based on our eligibility criteria. The summation of participants, study design, and type of interventions, duration of intervention and effectiveness of the intervention were presented using tables and figures.

The interventions found in the review varied significantly in terms of type, duration, and delivery methods. This heterogeneity presents challenges in making direct comparisons and using meta-analysis to draw definitive conclusions about the most effective interventions. Therefore, we narrated the findings of each study to draw conclusions. Future review should focus on individual intervention and its effectiveness to analyze meta-analysis to size of effect.

## 3. Result

The systematic review found 520 articles through the databases Scopus, Pub-Med/Medline, and Google Scholar. Of the 520 articles found, 13 that met our eligibility criteria were included in our systematic review. The number of articles that were excluded in each stage was illustrated in the systematic review reported to the Preferred Review Items for Systematic Review and Meta-analysis (PRISMA-2020) in Fig 1. The findings were presented in a structured manner, highlighting interventions, durations of intervention in Sub-Saharan Africa, and their impact on health outcomes.

### 3.1 Studies characteristics

Of 13 studies included in this review, 30.8% (4) were from South Africa [28–32], 4 studies were from Ethiopia [33–36], and other countries such Rwanda [37], Cameroon [38], Kenya [39], and Ghana [40]. Additionally, study was conducted across both South Africa and Malawi [21].

Of these studies, 92.3% (12) had randomized control trials (RCT) designs, while one study was quasi-experimental (QE), involving both interventional and control groups [34]. The total number of adults with diabetes who participated in both intervention and control groups vary across studies, providing insights into the scale and scope of this research. A total of 3171 adults with diabetes participated in both interventional and control groups. In the intervention group, a total of 1601 (50.5%) individuals attended, whereas in the control groups 1571 individuals with diabetes attended. The mean age across ten studies of intervention and control groups was 51.7 years. The remaining three studies did not reported age (Table 1).

### 3.2 Intervention and its implementation

In our systematic review, we identified three groups of interventions in Sub-Saharan African countries. These include diabetes self-management education, phone-based short messaging service (SMS) text messaging and other specific interventions.

Of 13 studies included in this review, different types of interventions were employed.

Five studies used nurse led diabetes self-management education intervention. These studies used interventions diabetes self-management education by the nurses' professionals on various

PRISMA 2020 flow diagram for new systematic review which included search databases and registers only

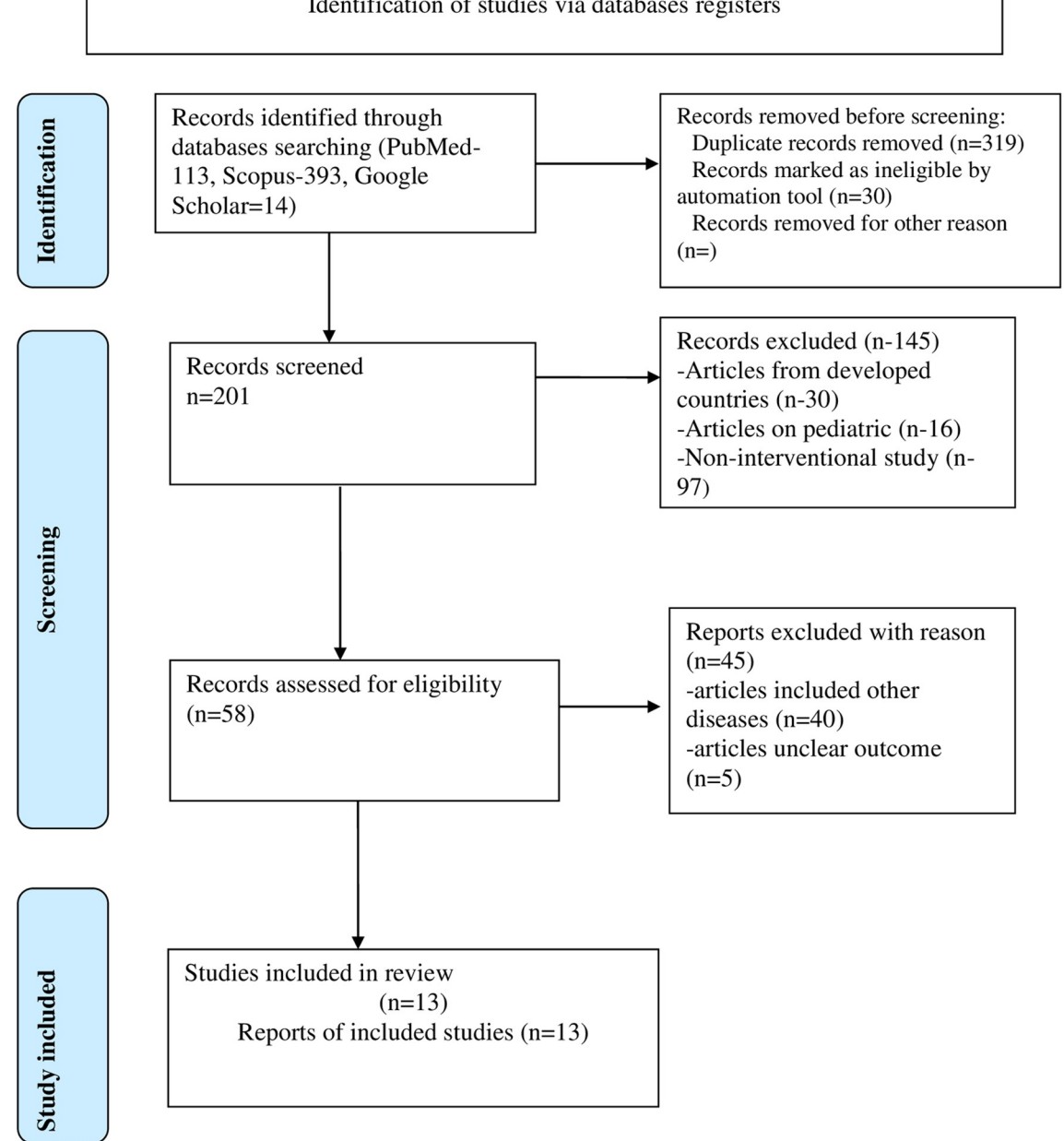

**Fig 1. PRISMA diagram.**

aspects of diabetes management including medication adherence, dietary habits, physical activities, glucose self-monitoring, and foot care, and overall care of behaviors.

Three studies employed SMS text messaging interventions for diabetes treatment adherence. SMS text messaging interventions were sent to patients with diabetes to remember to take medication, and glucose testing. Other specific interventions were family supported diabetes self-management education, glucose machine provision with training, and periodontal treatment of 10% iodine and diabetes education.

**Table 1. Characteristics of the eligible studies.**

|  | Category (n-13) | Frequency | Percentage (%) |
|---|---|---|---|
| Country/Area | South Africa | 4 | 30.8 |
|  | Ethiopia | 4 | 30.8 |
|  | Other countries | 5 | 38.4 |
| Study design | RCT | 12 | 92.3 |
|  | QE | 1 | 7.7 |
| Number of participants in the intervention group |  | 1601 | 50.5 |
| Number of participants in the control group |  | 1571 | 49.5 |
| Mean age of participants in years | (n = 10) | 51.7 |  |

Regarding the duration of intervention, 61.5% of the studies had duration of 6 months or less. The average follow-up time was 6.8 months, with durations ranging from a minimum of 2 months to a maximum of 12 months (Table 2).

## 3.3 Intervention and its effectiveness

Of the 13 studies included in our systematic review, 38.5% were effective in either of the primary or secondary outcomes measured. A few studies that showed positive changes [33–35,37,38], and while others did not show any positive change in outcomes illustrated blown (Table 3).

**3.3.1 Diabetes self-management education.** The first group of interventions focused on diabetes self-management education for the intervention group, whereas the control groups received usual care. Among 13 studies reviewed, seven studies were employed diabetes self-management education, with approximately only two showing positive changes in diabetes self-care knowledge and primary or secondary outcomes measured [33,34] (Table 3).

**3.3.2 Phone-based text SMS messaging intervention.** Three studies comprised this group, employing SMS text messaging for the intervention group on daily and weekly basis,

**Table 2. Intervention implementation, and duration of intervention on diabetes self-care interventions.**

| No | Study | Intervention and its implementation | Duration in months |
|---|---|---|---|
| 1 | [37] | Provided glucose machine and training, and the control group received usual care | 6 |
| 2 | [36] | Nurse-led diabetes self-management education to improve self-managing practices and behaviors | 6 |
| 3 | [29] | Diabetes self-management education on glycemic and metabolic control | 6 |
| 4 | [39] | Diabetes self-management education associated with metabolic management, complication prevention, and active screening. | 12 |
| 5 | [38] | Periodontal treatment with 10% Iodine and diabetes education for the intervention group and routine care for the control group | 3 |
| 6 | [33] | Diabetes self-management education by nurse professionals | 9 |
| 7 | [21] | SMS text messaging intervention to improve diabetes treatment adherence | 3 |
| 8 | [30] | SMS text Messaging intervention to improve outcomes of glycemia and medication adherence | 12 |
| 9 | [40] | Diabetes self-management education intervention by nurses | 3 |
| 10 | [31] | SMS text Messages intervention | 6 |
| 11 | [34] | Diabetes self-management education by nurses | 6 |
| 12 | [35] | Family-supported diabetes self-management education | 2 |
| 13 | [32] | Nutritional education by a nutritionist | 12 |

**Table 3. Interventions, outcomes measured, key findings, and effectiveness of diabetes self-care intervention strategies and their effectiveness in Sub-Saharan Africa.**

| Study | Intervention | Primary outcomes | Secondary outcomes | Key findings | Effectiveness of the intervention |
|---|---|---|---|---|---|
| [37] | Provision of glucose machine and training | Adherence to blood glucose testing and change in HbA1c levels | - | 64.3% of adherence to blood glucose testing and reduction of HbA1c 7.96 to 7.07% among intervention group | Effective |
| [36] | Nurse-led Diabetes Self-Management Education | Psychosocial health, eye examination | - | No significant differences in psychosocial health intervention and control group, but increased eye examination frequency was observed in the intervention group. | Not effective |
| [29] | Diabetes Self-Management Education | HbA1c | Diabetic-related complications | No significant change in HbA1c outcome, before and after education, but improved detection of diabetic-related complications among intervention group. | Not effective |
| [39] | Diabetes Self-Management Education | HbA1c | Blood pressure and body mass index | No significant change in HbA1c outcome or metabolic control in intervention group after education given. | Not effective |
| [38] | Periodontal treatment of 10% Iodine and diabetes education | HbA1c | - | Significant, 2.2% reduction in HbA1c levels in the group after periodontal treatment in intervention group. | Effective |
| [33] | Diabetes Self-Management Education | Diabetes knowledge, self-care behaviors | Practices of diet, exercise, glucose self-monitoring, and foot care | Significant improvement in diabetes knowledge scores among intervention group from 10.41 to 11.33 and 3.91 to 5.06 adherence general diet, and 5.07-to-5.80-foot care. However, there is no change in diabetes self-care behavior regimens or diabetes self-efficacy. | Effective |
| [21] | SMS text messaging intervention | HbA1c | Factors influencing participants' engagement in text messaging | Sending SMS messaging for intervention group was not effective to bring reduction in HbA1c. Due to participants' limited ability to act on messages, there is limited impact on health outcomes. | Not effective |
| [30] | SMS text messaging intervention | HbA1c | | No significant effect on glycemic control after sending SMS messaging to intervention group, but a slight impact on blood pressure. | Not effective |
| [40] | Diabetes self-management education intervention | HbA1c | | Diabetes education for intervention group was not significant on glycemic control after education. | Not effective |
| [31] | SMS text messaging intervention | Blood glucose, blood pressure, and anthropometric parameters | Weight and BMI | The SMS text messaging was acceptable and feasible among the participants. However, its efficacy in improving glycemic status and other clinical outcomes not effective. | Not effective |
| [34] | Diabetes self-management education | Self-care knowledge and behaviors | - | Significant improvement in self-care knowledge and behaviors. The level of knowledge for the interventional group that at the baseline high level increased from 11.8%-54% and the low level decreased from 62.7% to 20.6%. Behavior at good was 15.7%-76.4%. | Effective |
| [35] | Family-supported diabetes self-management education | HbA1c | Blood pressure, BMI, and the Lipid profile | Significant improvement in HbA1c decreased by 1.1% and triglycerides. However, DSMES had no significant effect on blood pressure, BMI, total cholesterol, low-density, and high lipoprotein. Culturally tailored, social cognitive theory-guided, family-supported, community-based DSME program could benefit HbA1c and triglycerides. | Effective |
| [32] | Nutritional education | HbA1c | Blood pressure, BMI, lipid profile, and dietary behaviors | Nutritional education was not effective on HbA1c, blood pressure, BMI, and intake of macronutrients, vegetables, and fruits. However, goal setting and self-efficacy training could help future nutrition education for patients in resource-limited settings. | Not effective |
| | Effectiveness | | | | (5/13) = 38.5% |

aimed at treatment adherence, while the control group received usual care. However, these interventions did not demonstrate positive changes in either the primary or secondary outcomes measured [21,30,31] (Table 3).

**3.3.3 Other specific interventions.**   Among other specific interventions utilized, Nganga et al. employed a glucose testing machine with training and additional resources to assess adherence to glucose testing and change in HbA1c. This study exhibited a significant improvement in blood glucose testing adherence, reaching 64.3%, and reduction of 0.9% in HbA1c within the intervention group [37].

Another interventional study conducted in Cameroon utilized periodontal treatment with a 10% povidone iodine solution, demonstrating a positive outcome with a notable 2.2% reduction in HbA1c after a 3-month follow up [38]. Additionally, Diriba et.al utilized family support approach with culturally tailored education. The primary outcome focused on HbA1c, with secondary outcomes including blood pressure, BMI (body mass index), lipid profile, and dietary behaviors. This study observed significant improvement in HbA1c and triglycerides, though no effect was observed on other parameters [35] (Table 3).

## 4. Discussion

Achieving optimal diabetes treatment goals remains challenging for adults with type 2 diabetes in Africa, with complications often arising from inadequate self-care, particularly Sub-Saharan region [41]. This systematic review assessed diabetes self-care intervention strategies and their effectiveness in this region.

The identified intervention strategies fell into three categories: diabetes self-management education, phone-based SMS text messaging, and other specific interventions such as family-supported culturally tailored, glucose testing machine provision, and periodontal treatment with 10% iodine. The overall effectiveness of diabetes self-care intervention strategies in Sub-Saharan Africa, as in our review, was found to be 38.5% in either primary or secondary outcomes measured.

Diabetes self-management education intervention, delivered by nurse professionals, trained health promoters, and diabetes educators, showed promising but mixed results in our review. Evidence aligned with previous studies conducted in the WHO Africa Region and low-middle-income countries [42,43]. Other systematic review on diabetes self-management education in both developed and developing countries revealed more effectiveness than our study [44]. This might be because our study included only Sub-Saharan Africa, in which the participants had both financial constraints and limited education to implement diabetes self-management education [45]. In our review, diabetes self-management education intervention was not effective in glycemic control (HbA1c) outcomes. However, evidence in other studies showed a positive impact on HbA1c levels [23,46].

Phone-based SMS text messaging interventions, lasting from 3 to 12 months, were thought feasible and acceptable to participants. However, none demonstrated effectiveness in either in the primary or secondary outcomes measured. This is supported by other similar studies conducted in developing countries, where SMS text messaging was implemented for durations exceeding a year [47–49] and previous systematic review conducted by Linda et.al [50].

However, our review findings on SMS text messaging contrast with a study conducted in Egypt, where SMS text messaging intervention showed a significant effect on reducing HbA1 levels by 1% and fostering self-management behavior in Egypt [51]. This discrepancy might be attributed to the comparatively better socioeconomic status of the study participants in Egypt compared to those in Sub-Saharan Africa [52].

In the present review, family-supported, culturally tailored interventions, especially when combined with education, demonstrated effective outcomes in reduction HbA1c by 1.1% among intervention group [35]. This finding is in line with the conclusions drawn from previous research, such as the systematic review conducted by Islami [53], along others studies conducted in low-income Latinos [54], and within the Southeast Asia region [55]. These findings collectively underscore the vital role of family support in effectively managing diabetes self-care and reducing complications associated with the condition. Family involvement in healthcare, especially diabetes can significantly influence an individuals' ability to adhere to treatment plans and make necessary lifestyle changes [56]. Culturally tailored interventions that consider the unique cultural backgrounds and preferences of affected individuals and their families, tend to yield better results in terms of engagement and sustainable outcomes [57]. Moreover, the implications of these findings emphasize the need for healthcare providers and policymakers to recognize and integrate family-centered approaches into diabetes care strategies.

In this review, we observed increased in the adherence to self-monitoring of blood glucose up to 64.3% and reduction of mean HbA1c 7.96 to 7.07% among patients following the provision of glucose meters, training sessions, logbooks for record-keeping, and proper disposal mechanisms [37]. This finding aligns with findings from Parsons et al. [58] who reported similar outcomes in other resource limited settings, reinforcing the effectiveness of this intervention across diverse settings.

A particularly specific intervention, we review was the incorporation of periodontal treatment regime using a 10% iodine solution in conjunction with diabetes education programs significantly reduced 2.2% in HbA1c levels in the intervention group after periodontal treatment[59]. This integrated approach not only addresses oral health, which is often neglected in diabetes management, but also reinforces the importance of holistic treatment frameworks in proving overall health outcomes for diabetic patients in this region. This outcome is similar to the findings of the studies conducted in various global regions [60,61], which indicated that non-surgical periodontal treatment effectively contributes to managing diabetes self-care while reducing complications associated with periodontal health in diabetes patients. This showed that the management of diabetes involves a multifaceted approach beyond medication. Periodontal health plays a significant role, as studies have consistently shown a bidirectional relationship between glycemic control and periodontitis [62,63].

## 5. Strengths and limitations the study

### 5.1 Strengths

The review covered various aspects of diabetes self-care, including education, phone-based SMS text interventions, family involvement, equipment provisions and periodontal health. This breadth of coverage allows for a comprehensive evaluation of different aspects of diabetes management.

The review focused on the randomized control trials, which are considered the gold standard in assessing intervention effectiveness. This methodology enhances the reliability of the findings and strengthens the review's credibility.

The review considered a diverse range of interventions, durations, and outcomes, providing a comprehensive overview of diabetes self-care strategies in Sub-Saharan Africa. This inclusivity allows for a more holistic understanding of the interventions' effectiveness.

With focus on Sub-Saharan Africa, the review provides contextually relevant insights into interventions tailored to the socio-economic and cultural landscapes of this region. This targeted approach contributes to the literature on diabetes management in resource-constrained settings.

### 5.2 Limitations

The interventions assessed in the review varied significantly in terms of type, duration, and implementation approaches. Such heterogeneity presents challenges in making direct comparisons and might limit the ability to draw definitive conclusions about the most effective interventions.

The review included a relatively small number of studies (13) that met the eligibility criteria. A large pool of studies would have provided more significant statistical power and possibly stronger conclusions.

There is a possibility of bias towards published studies with positive outcomes, potentially overlooked unpublished or negative findings. Our studies also limited only English language, there excluding studies in other languages. This bias could influence the overall perception of intervention effectiveness.

The variations in outcomes measures across studies, such as different interventions for glycemic control could make it difficult to compare and synthesize results uniformly. Therefore, we did not conduct pooled data analysis and effect of size of each study's outcome.

However, despite these limitations, the review provides valuable insights into the setting of diabetes management in the region. It also underscores an opportunity for further investigations and improvement in healthcare practices.

## 6. Conclusion and recommendation

Our systematic review revealed a range of diabetes self-care intervention strategies in Sub-Saharan Africa. These intervention strategies demonstrated varying levels of effectiveness in improving glycemic control and self-care practices. Particularly, interventions that were culturally tailored and supported by family involvement, equipment provision, were more effective in enhancing adherence and glycemic control. However, test messaging intervention did not significantly improve outcomes, highlighting a need for further research to tailor technological solutions to local needs. For effective diabetes management in the region, a comprehensive approach that incorporates education, family support, and access to glucose testing devices is crucial. Future research should focus on robust study designs and expanding geographical coverage to better understand and enhance the efficacy of intervention strategies for self-care in Sub-Saharan Africa.

## Supporting information

**S1 Fig. PRISMA checklist.**
(DOCX)

**S1 File. Supplementary 1 quality checklist.**
(DOCX)

**S2 File. Supplementary 2 included studies.**
(XLSX)

## Acknowledgments

The author extends the heartfelt appreciation for Walailak University for PhD. Scholarship, and Wolaita Sodo University College of Medicine and Health Sciences for supporting my family. My especial thanks goes to the School of Public Health for making a conducive environment for my study such as internet services, AC monitored room, and other necessary supports.

## Author Contributions

**Conceptualization:** Temesgen Anjulo Ageru, Cua Ngoc Le, Apichai Wattanapisit, Eskinder Wolka Woticha, Charuai Suwanbamrung.

**Data curation:** Temesgen Anjulo Ageru, Cua Ngoc Le, Apichai Wattanapisit, Eskinder Wolka Woticha, Muhammad Haroon Stanikzai, Charuai Suwanbamrung.

**Formal analysis:** Temesgen Anjulo Ageru, Nam Thanh Truong, Muhammad Haroon Stanikzai, Temesgen Lera Abiso, Charuai Suwanbamrung.

**Investigation:** Eskinder Wolka Woticha, Muhammad Haroon Stanikzai.

**Methodology:** Temesgen Anjulo Ageru, Apichai Wattanapisit, Eskinder Wolka Woticha, Nam Thanh Truong, Muhammad Haroon Stanikzai, Temesgen Lera Abiso, Charuai Suwanbamrung.

**Project administration:** Charuai Suwanbamrung.

**Resources:** Charuai Suwanbamrung.

**Software:** Nam Thanh Truong, Temesgen Lera Abiso.

**Supervision:** Temesgen Lera Abiso.

**Validation:** Cua Ngoc Le, Apichai Wattanapisit.

**Visualization:** Cua Ngoc Le, Eskinder Wolka Woticha.

**Writing – original draft:** Temesgen Anjulo Ageru, Apichai Wattanapisit, Eskinder Wolka Woticha, Muhammad Haroon Stanikzai, Charuai Suwanbamrung.

**Writing – review & editing:** Temesgen Anjulo Ageru, Cua Ngoc Le, Muhammad Haroon Stanikzai.

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
