## [Decision Letter · Decision Letter 0]

1 May 2024

PONE-D-24-05477Diabetes self-care intervention strategies and their effectiveness in Sub-Saharan Africa: A systematic reviewPLOS ONE

Dear Dr. Suwanbamrung,

Thank you for submitting your manuscript to PLOS ONE. After careful consideration, we feel that it has merit but does not fully meet PLOS ONE’s publication criteria as it currently stands. Therefore, we invite you to submit a revised version of the manuscript that addresses the points raised during the review process.

We look forward to receiving your revised manuscript.

Kind regards,

Chikezie Hart Onwukwe

Academic Editor

PLOS ONE

Journal Requirements:

3. Please remove your figures from within your manuscript file, leaving only the individual TIFF/EPS image files, uploaded separately. These will be automatically included in the reviewers’ PDF.

Reviewers' comments:

Reviewer's Responses to Questions

**Comments to the Author**

1. Is the manuscript technically sound, and do the data support the conclusions?

Reviewer #1: Partly

Reviewer #2: Yes

2. Has the statistical analysis been performed appropriately and rigorously? 

Reviewer #1: N/A

Reviewer #2: N/A

3. Have the authors made all data underlying the findings in their manuscript fully available?

Reviewer #1: No

Reviewer #2: Yes

4. Is the manuscript presented in an intelligible fashion and written in standard English?

Reviewer #1: Yes

Reviewer #2: Yes

5. Review Comments to the Author

Reviewer #1: This study does not provide the most important information "How the SSA was included". In the search term the SSA does not exist. Did the authors search for a particular authors or not?

please provide some information about the level of literacy of the population. Des this affects the resulsts. Did the authors revise only abstract or full manuscriots.

Reviewer #2: • The introduction section was good and from mz side there is nothing toadd

• It seems that you are here assuming from the beginning that there will be a reduction in complication:

"Outcomes included glycemic control, knowledge and behavioral change, adherence, and reduced complications."

• It is not clear if you have chosen these inclusion criteria while you were selecting the papers or you are just mentioning here what you have after you finished the screening. If you have selected these criteria from the beginning, then please justify using these outcomes.

"Outcomes included glycemic control, knowledge and behavioral change, adherence, and reduced complications."

• Why was the time frame set to be from 2013 and not before?

• It is not clear how adherence could be among the outcomes and intervention:

The chosen interventions were specifically centered around diabetes self-care, with particular emphasis on aspects such as diet, glucose monitoring, medication adherence, physical activities, and psychosocial support. "Outcomes included glycemic control, knowledge and behavioral change, adherence, and reduced complications."

• In Data analysis, the parameters used to describe the data were not reported.

• Using the year of publication as a factor to categorize the data seems irrelevant and adds no important information to the paper.

• Reporting the origin of the studies in the first section of the results is not needed. You can mention the two countries with the highest number of studies. Avoid repetition of data from the table.

• The age of the participants (mean or range) is very important and relevant to the paper.

• The interventions were defined differently in the inclusion criteria and in the results. In the inclusion criteria: "The chosen interventions were specifically centered around diabetes self-care, with particular emphasis on aspects such as diet, glucose monitoring, medication adherence, physical activities, and psychosocial support." In the results: "In our systematic review, we identified three groups of interventions in Sub-Saharan African countries. These include diabetes self-management education provided by nurses and other healthcare providers, phone-based short messaging service (SMS) text messaging, and other specific interventions."

• There are no specific numbers used to describe the interventions' effectiveness. The comparison with the control group is unclear, and the significance of the difference was not reported. Using general terms to describe the effectiveness is not sufficient. Providing numbers for the control and intervention groups with relevant significance is essential to convey your point.

• Table 3 should contain the number of participants in the control and intervention groups for each study, the effectiveness for each primary and secondary outcome, and the significance of results. That would make the findings clearer instead of using general terms.

• There is repetition in the method section: "Thirteen eligible studies published between 2013 and 2023 were reviewed. The review revealed diverse interventions, implementation strategies, durations, and their effectiveness in improving diabetes management within the region." The first section in the discussion should only summarize the results.

• Repetition of citations is unnecessary in the discussion (Second discussion part). Instead, report the strategies found in your research without re-mentioning the citations.

• Avoid using general terms in the discussion. Instead, provide specific numbers to reflect the significance of the intervention's efficacy. For example, instead of saying, "a significant effect on reducing HbA1 levels," specify how much HbA1c was reduced and whether it was clinically relevant.

• The discussion contains many general terms that should be specified using numbers to make the argument specific and scientifically precise, such as:

- "This finding is consistent with a similar study conducted in another region of Africa."

- "However, our findings contrast with a study conducted in Egypt, where a 3-month SMS text messaging intervention showed a significant effect on reducing HbA1 levels and fostering self-management behavior."

- "Family involvement in healthcare, especially diabetes, can significantly influence an individual's ability to adhere to treatment plans and make necessary lifestyle changes."

• Delete repetitions in the paper to make the discussion robust and of high quality, such as:

- "In our review, the provision of a glucose machine, along with training, logbooks for registration, and waste management materials, significantly enhanced the adherence to self-monitoring of blood glucose after a 6-month follow-up."

- "The present systematic review demonstrated a notably high adherence to glucose self-monitoring due to the provision of glucose testing machines and the comprehensive training and support provided to the participants. This underscores the significance of supporting patients with diabetes, as it significantly improves adherence and consequently reduces complications related to diabetes."

• The conclusion is too long and scould be summerized

6. PLOS authors have the option to publish the peer review history of their article (what does this mean?). If published, this will include your full peer review and any attached files.

Reviewer #1: No

Reviewer #2: **Yes: **Ahmad Altom

---

## [Author Response · Author response to Decision Letter 0]

27 May 2024

We have attached a response letter.

---

## [Decision Letter · Decision Letter 1]

6 Jun 2024

Diabetes self-care intervention strategies and their effectiveness in Sub-Saharan Africa: A systematic review

PONE-D-24-05477R1

Dear Dr. Charuai Suwanbamrung,

We’re pleased to inform you that your manuscript has been judged scientifically suitable for publication and will be formally accepted for publication once it meets all outstanding technical requirements.

Kind regards,

Chikezie Hart Onwukwe

Academic Editor

PLOS ONE

Additional Editor Comments (optional):

Reviewers' comments:

Reviewer's Responses to Questions

**Comments to the Author**

1. If the authors have adequately addressed your comments raised in a previous round of review and you feel that this manuscript is now acceptable for publication, you may indicate that here to bypass the “Comments to the Author” section, enter your conflict of interest statement in the “Confidential to Editor” section, and submit your "Accept" recommendation.

Reviewer #1: All comments have been addressed

2. Is the manuscript technically sound, and do the data support the conclusions?

Reviewer #1: Yes

3. Has the statistical analysis been performed appropriately and rigorously? 

Reviewer #1: Yes

4. Have the authors made all data underlying the findings in their manuscript fully available?

Reviewer #1: Yes

5. Is the manuscript presented in an intelligible fashion and written in standard English?

Reviewer #1: Yes

6. Review Comments to the Author

Reviewer #1: Dear Authors, thank you for the revised version of the manuscript.

You respond to all my queries and I have no additional questions to add.

7. PLOS authors have the option to publish the peer review history of their article (what does this mean?). If published, this will include your full peer review and any attached files.

Reviewer #1: **Yes: **Guenka Petrova

---

## [Editor Report · Acceptance letter]

19 Jun 2024

PONE-D-24-05477R1 

PLOS ONE

Dear Dr. Suwanbamrung, 

I'm pleased to inform you that your manuscript has been deemed suitable for publication in PLOS ONE. Congratulations! Your manuscript is now being handed over to our production team.

Kind regards, 

on behalf of

Dr. Chikezie Hart Onwukwe 

Academic Editor

PLOS ONE